# Extracting Narrative Patterns in Different Textual Genres: A Multilevel Feature Discourse Analysis

**María Miró Maestre *** , **Marta Vicente** , **Elena Lloret** and **Armando Suárez Cueto**

Department of Software and Computing Systems, University of Alicante, San Vicente del Raspeig, 03690 Alicante, Spain
* Correspondence: maria.miro@ua.es

**Abstract:** We present a data-driven approach to discover and extract patterns in textual genres with the aim of identifying whether there is an interesting variation of linguistic features among different narrative genres depending on their respective communicative purposes. We want to achieve this goal by performing a multilevel discourse analysis according to (1) the type of feature studied (shallow, syntactic, semantic, and discourse-related); (2) the texts at a document level; and (3) the textual genres of news, reviews, and children's tales. To accomplish this, several corpora from the three textual genres were gathered from different sources to ensure a heterogeneous representation, paying attention to the presence and frequency of a series of features extracted with computational tools. This deep analysis aims at obtaining more detailed knowledge of the different linguistic phenomena that directly shape each of the genres included in the study, therefore showing the particularities that make them be considered as individual genres but also comprise them inside the narrative typology. The findings suggest that this type of multilevel linguistic analysis could be of great help for areas of research within natural language processing such as computational narratology, as they allow a better understanding of the fundamental features that define each genre and its communicative purpose. Likewise, this approach could also boost the creation of more consistent automatic story generation tools in areas of language generation.

**Keywords:** natural language processing; discourse analysis; corpus linguistics; computational linguistics; communicative objectives



## 1. Introduction

Automatic understanding and generation of narratives represent two complementary tasks of a very prolific area of research addressed as computational narratology or narrative intelligence. Computational narratology aims at facilitating annotations and supporting literary scientists with their analysis [1]. It comprises the study of different elements from narratives such as characters, temporal dimensions, event composition, and ordering to explore and test literary hypotheses through mining those narrative structures from corpora [2].

In this sense, we understand narrative as a textual typology that "primarily introduces events and states entities into the universe of discourse. They are temporally related to each other; tense conveys continuity" [3]. Narrative shows a long tradition of scientific research, where authors from different disciplines place their interest in deciphering some of the natural components of this textual typology applied to several areas of study such as philosophy, psychology, education, etc. Indeed, narratives generally share a basic structure and a set of linguistic features, but they can also contain particular elements that may not appear as a general rule in texts belonging to this typology. However, those genres which show different elements or surface structures can still be classified as narratives as long as their semantic interpretation is equivalent [4].

With this in mind, the motivation for the present study is to define to a certain extent how the structure of three types of narratives can be formalized, as well as to detect and extract their discriminating features from their automated processing. This idea is in line with the results of some authors who also focused their research on this area, as in the case of Igl and Zeman [5], who agreed that narrative "cannot be defined by one single criterion but rather refers to a category that shares a variety of characteristic prototypical features and has to be seen as 'a fuzzy set allowing variable degrees of membership'" ([6], similarly also Herman [7,8]). Indeed, in order to analyze the narrative texts included in the following section, a selection of corpora from different genres with concrete communicative goals was carried out. This approach was inspired by much of the work devoted to genre study [9,10], where purpose is used as an often primary criterion for deciding whether a given discourse falls within a particular generic category [11]. Following Swales [9], the principal feature that turns a collection of communicative events into a genre is some shared set of communicative purposes, therefore becoming the basic linguistic element that will help us to identify which linguistic features tend to shape each genre. In this manner, our corpora are composed of news, children's tales, and user-generated reviews, which are illustrative cases of three communicative goals, i.e., to inform, to entertain, and to persuade. These genres differ in terms of purpose, but also share several patterns, all of them being examples of narrative discourse. Thus, the present multigenre analysis could be suitable to support the characterization of what they have in common and what distinguishes them.

Consequently, this research aims to confirm if specific narrative features are equally represented in different genres by comparing their prevalence in the documents that form the corpora compiled for each genre. To adequately perform such comparative linguistic analysis, we need to address the following research questions:

- (RQ1) Which linguistic features seem to be generally linked to each textual genre, and how prevalent are they?
- (RQ2) Is it possible to establish a connection between particular linguistic features and certain genres, given their main communicative purposes?

This type of analysis could have an impact on different tasks in the scientific area of computational linguistics. In particular, a better comprehension of the structure of the elements in the discourse of each genre will help natural language processing (NLP) tasks to deal with storylines and narrative linguistic patterns from a more informed perspective, improving understanding and generation tools. Question answering, textual entailment, or automated journalism are some of the tasks that could take advantage of this approach in their development, thanks to the definition of which features are commonly linked to each genre to recognize them as such.

The paper is organized as follows. We first introduce the related work on narratives and their statistical feature analysis in Section 2. Then, Section 3 describes the corpora collected together with the annotation tools used. The methodology to extract and prepare the data from each corpus is detailed in this section too. Next, we provide the multilevel linguistic analysis of the results in Section 4. Then, Section 5 serves as a brief discussion of the findings described in the previous analysis and, finally, Section 6 summarizes our conclusions and future work that could be developed based on this paper.

## 2. Related Work

In linguistic theory, defining the most representative features from narrative texts across several genres still poses a challenge that brings together the efforts of multiple tasks and disciplines. These days, linguists [12], political scientists [13], psychologists [14], journalists [15], and computer scientists [16] are becoming involved in better understanding stories and their role in our experiences. Aiming to contribute to this movement, our starting point is a conception of discourse which stipulates that the genre to which it belongs shapes the expression and structure that underlies it. However, these new perspectives on the linguistic analysis of narrative have also motivated the emergence of various concepts with a great impact on genres. Following Bhatia [17], one of them is the notion of interdiscursivity,

"whose emergence has been triggered by the growing awareness of the need to see genres in relation to other genres with which they are interrelated in various ways, in contrast with the traditional view of genres as clearly distinct and discrete entities" [18]. As a consequence, we agree with De Fina and Georgakopoulou [12] that the definitions of narrative should not be genre-specific or genre-discriminating.

Due to this "interdiscursive" conception of genres, many researchers developed previous studies on narrative in order to show results applicable to a wider variety of genres. Indeed, although some of these ideas were already stated in traditional narrative theory, the first publications on the structural and conceptual study of this textual typology were focused on the distinction of its meaningful parts. According to Labov and Waletzky [4], a narrative consists of at least three sections: orientation, complicating action, and evaluation. This conception was present, for example, in the work of Ouyang and McKeown [19] and Swanson et al. [20]. Another approach largely adapted throughout the years in this area of research is the one based on the Proppian morphology of narratives [21]. Propp argued that a narrative was constituted by a sequence of what he called *functions*.

He defined a maximum of 33 types of functions that could occur in a story. Although his work initially focused on the specific domain of folk tales, many studies have been influenced by his theories [22,23].

As research increased in this area of linguistics, several authors started to adapt this narrative theory to wider scopes to discover more characteristics that defined different textual genres. Regarding Biber [24,25], Biber and Conrad [26], he was the author to introduce the research approach called "multidimensional (MD) analysis" of genres and registers. In his work, Biber used computer-based text corpora and computational tools to identify linguistic features in texts, and multivariate statistical techniques to analyze the co-occurrence relations among linguistic features, thereby identifying underlying dimensions of variation in a language [27]. Indeed, his work has largely influenced this area of linguistic analysis, fostering further research with this methodology on corpora extracted from various resources such as Twitter [28] or university assignments [29], and even applying it to other languages such as Russian [30].

Similarly to Biber, this study aims to perform such linguistic analysis using statistical techniques as well, but distributing the selected features in the different analysis levels of language. This approach was recently used in studies of the same linguistic area, as in the case of Sung et al. [31]. In their study, the authors applied the multilevel analysis (including word, syntax, semantics, and cohesion) to the Chinese language to create models that improve the readability of texts through machine learning. Another example is that of Qiu et al. [32], who established four categories of linguistic metrics, including shallow, POS, syntactic, and discourse features, to define their relations within documents for readability classification. Regarding Cimino et al. [33], the authors created a four-folded typology of features automatically extracted from texts pertaining to four genres to investigate the task of automatic genre classification, showing the significant role of syntactic features when classifying those texts.

In the present study, the three genres selected—news, reviews, and children's tales—share several patterns reflected in their linguistic features that make them be considered as narratives. Such narrative identity is arguably due to their tendency to include passages focused on the progression of an action based on events. Children's tales are the most prototypical example, as their main purpose is to narrate a story chronologically. Regarding news and reviews, they also show this narrativity, with news developing the succession of events within a particular topic of current interest, and reviews thanks to online users who become highly involved in narrating their own experiences with products or services.

This identification of genre patterns belonging to a particular text typology has been largely studied by many authors who have devoted their research to a general categorization of those features that shape genres individually. This is the case of Zen [34], who focused on an analysis of the language used in children's literature by four well-known

English writers. She based her study on corpus linguistics to show the features used by these four authors that make this genre a very distinguished text in comparison to adult literature. Another approach is that of Hansen [15], who also made use of corpus linguistics to analyze future-oriented or unreal news by focusing on Danish articles with political themes that show a growing speculative intention. Fostered by the rise of online linguistic genres, we find the approach of Almiron-Chamadoira [35]. In the article, the author showed a journalist's perspective on the review genre of the online platform Amazon, collecting 61 features belonging to several levels of analysis such as rhetoric and structure. Her findings reflect a mixture of particular patterns linked to informative discourse, as well as specific linguistic features defining this genre.

Arguably, scientific research in this area shows a tendency towards an individual analysis of genres pertaining to the narrative typology and based on generic features that do not delve into specific properties of those characteristics or comparisons between various genres. As a consequence, these previous studies inspired us to develop a data-based approach according to some of the aforementioned linguistic theories. In this manner, the main aim of the present research is to extend this type of analysis to several genres with a greater compilation of features across several linguistic levels. We believe that this gap in comparative narrative research could give computational linguistics the opportunity to bring to light more accurate results that may benefit the study of each interlinked linguistic level, as well as show further insights into the characterization of those genres.

## 3. Data and Tools

For the purpose of the present study, several linguistic compilation and processing steps need to be performed first to appropriately address the multilevel linguistic discourse analysis of the different features belonging to news, reviews, and children's tales. Consequently, following a data-based approach [36–38], the next subsections provide further information on the computational tools employed in the sets of corpora from each genre. In this way, we are able to extract the statistical results of the previously defined features that serve as the basis of the subsequent multilevel feature discourse analysis.

### 3.1. Corpora Collection

Three textual genres were selected to complete the present comparative analysis: news, reviews, and children's tales. As was already mentioned, each genre can be identified as a type of narrative given their communicative purposes; however, they still show different linguistic features that cause them to be considered as separate genres, *per se*. In line with this, several corpora were collected from various sources in English to compile a balanced proportion of texts that reflected the most prevalent features in each genre. With respect to children's tales, we gathered stories automatically from *Bedtime Stories* [39], together with the Lobo and Matos corpus of fairy tales [40]. Regarding news, the dataset comes from the Document Understanding Conference (DUC) [41], specifically DUC-2004. Finally, reviews were taken from the SFU corpus [42]. This corpus contains reviews from various types of products (cars, books, movies, etc.), bringing more variety in the style of the documents, therefore allowing us to contemplate a wide range of phenomena in the genre.

The statistics of the corpora are indicated in Table 1. Despite the different values that the total number of words and sentences show for each genre (with more than 300,000 words for reviews and tales and 274,000 for news), the collected corpora still serve as a balanced compilation of texts of each of the genres included for the present study. This is because the number of documents for each genre ranges from 447 to 496, and the mean length of sentences per document is similar in all of them, varying between 21 and 26 words. Consequently, both aspects make feasible the comparative research to be undertaken, given that our focus here is on the document level of each narrative genre.

**Table 1.** Description of the corpora.

|         | # Docs | # Sents | # Words | Sents/Doc | Words/Doc | Words/Sent |
|---------|--------|---------|---------|-----------|-----------|------------|
| News    | 487    | 12,565  | 274,153 | 26        | 563       | 26         |
| Reviews | 447    | 17,150  | 306,120 | 38        | 685       | 21         |
| Tales   | 496    | 16,236  | 344,606 | 33        | 695       | 24         |

*3.2. Linguistic Processing*

Taking into account that we aim to perform a multilevel feature discourse analysis, a set of linguistic elements, covering shallow, part-of-speech, syntactic, semantic, and discourse information, was first defined. This collection of features to be further analyzed was also in line with the type of information that can be obtained using NLP tools (as shown in [32,43,44]). Specifically, all the documents gathered for this research were processed with the following linguistic analyzers:

- Freeling [45], a popular multilingual tool that allows us to obtain lexical, syntactic, and semantic information from a document. For example, features such as the presence of types of phrases, specific grammatical elements, or named entities were obtained thanks to this tool.
- AllenNLP [46]. This tool was used for the particular task of coreference resolution, as AllenNLP currently represents most of the state of the art on this specific research topic. Indeed, Freeling also includes a coreference resolution module, but it was observed that AllenNLP gave more adequate and complete results for the purpose of the present study. The coreference resolution model used is a model based on [47].
- CAEVO (Cascading Event Ordering system) [48], a tool capable of extracting and classifying discursive information related to events, time, and temporal expressions. For this purpose, it takes into account the TimeML specification [49], according to which an *event* refers to something that occurs or happens, and can be articulated by different kinds of expressions such as verbs, nominalizations, or adjectives. In addition, the tool classifies events semantically into one of seven categories: aspectual, perception, state, reporting, intensional action, intensional state, and occurrence. With this tool it is possible to extract all the interesting information regarding the *event phenomena*, not only with the terms that the tool identifies as *events*, but also their semantic environment.

In addition to these NLP tools, we also made use of the *lexicon of prototypical discourse markers* [50] to identify such features across the documents, so that they could be subsequently used to show an argumentative representation of the text.

## 4. Multilevel Feature Study of the Genres

In order to answer the two research questions considered for the present study, a set of features has to be defined and extracted with the aforementioned computational tools. Then, a deep and individual analysis of each of them is conducted. To accomplish this, the linguistic features analyzed in the three genres were divided into four levels of linguistic analysis, i.e., shallow, syntactic, semantic, and discourse features. Such classification originates in the main function that each feature provides to the structure of the text at the document level. Following this data-based methodology, the average presence of certain elements along the different documents was measured. Thus, by means of this linguistic discourse analysis on several levels, we will be in a position to find how these elements influence the prototypical structure of the textual genres of news, reviews, and tales, and how these features help to make a distinction between the genres. Furthermore, considering the characteristics the three genres share, it will be also possible to detect which linguistic mechanisms are more common to narrative discourse, regardless of their individual prototypical features.

*4.1. Shallow Features*

Shallow features are generally used to quantify some simple aspects of the words and sentences that comprise each text, i.e., the average number of words in each sentence or the average number of sentences that make up a text [51]. For the present analysis, some of these linguistic features were tested to check how they affect the overall shape of each textual genre, including both the average word length of the words that comprise the texts and the average presence of commas within each document.

4.1.1. Word Length

We start the analysis by comparing the average word length in each of the three genres. As can be seen in Table 2, news tend to show words with a minimum of seven letters (with an average of 160.72 words per document) in comparison to tales, where the average number of letters in words oscillates between three (181.90) and four (142.56). Meanwhile, reviews show similar word lengths except for those made up of five and six letters, which are less frequent in the genre, with an average of 77.01 and 51.27, respectively.

**Table 2.** Average shallow features per document.

| Features | News | Reviews | Tales |
|---|---|---|---|
| Word length 1 | 77.37 | 123.79 | 129.11 |
| Word length 2 | 97.55 | 111.09 | 117.98 |
| Word length 3 | 105.07 | 145.62 | 181.90 |
| Word length 4 | 81.50 | 124.25 | 142.56 |
| Word length 5 | 55.36 | 77.01 | 78.78 |
| Word length 6 | 46.85 | 51.27 | 52.52 |
| Word length 7 | 160.72 | 130.85 | 95.94 |
| Punctuation, commas | 32.90 | 33.61 | 58.99 |

Focusing on tales, these findings can be linked to the target audience of the genre, which is mostly children. Indeed, apart from entertaining being their general communicative purpose, tales also contain an instructive component by which children are taught moral lessons [52]. Thus, to both entertain and learn, children need to understand almost all words that appear in the story so they can keep up with its development, therefore preferring short words mostly belonging to a child's lexical world [52]. The opposite findings are reflected within news, where the prevalence of words with a minimum of seven letters is due to their formal vocabulary and writing style [53]. Finally, reviews appear to be the most balanced genre between words of different lengths. Such variety can be linked to the subjective nature of the genre, as each user freely chooses the vocabulary of their review. However, the register generally preferred in this genre tends to be informal to show more proximity to the community of users so that they can relate to such experience [54]. Likewise, this balance of word length can also be due to the range of products or experiences to rate, as specific types of products within technical fields such as mechanics or technology may need to use longer technical terms in comparison to more generic products.

4.1.2. Commas

Orthographic elements can also influence the definition of textual genres. While periods are implicitly considered by linguistic tools to perform sentence segmentation, other elements that are closely related to narratives, such as quotation marks, are analyzed at their respective linguistic level, given their influence in the discourse.

Regarding commas, Table 2 shows how children's tales stand out as the genre with more presence of this orthographic feature (with 58.99 average commas per document), whereas news and reviews have almost equal results (32.90 and 33.61, respectively) with less prevalence in their respective corpora. Such contrast between children's tales and the other two genres may be due to one of the particular communicative intentions that writers pretend to accomplish for children's tales. In fact, as Guijarro [52] stated, tales have a clear

preference for revealing the information through simple sentences without elaborating on further explanations so that children do not become confused when distinguishing the different events that are taking place. As a consequence, commas become an excellent tool to help children recognize each new excerpt of the story inside a sentence, therefore facilitating children's comprehension by showing the events and actions that unfold the story progressively.

### 4.2. Part-of-Speech (POS) and Syntactic Features

According to Mishra and Bhattacharyya [55], syntactic features correspond to the grammatical structure of a language. Thanks to their analysis, we are able to check the predominant composition of the sentences included in each genre, as well as the type of words and syntactic relations that generally affect their overall structure.

#### 4.2.1. Nouns and Proper Nouns

Focusing first on nouns (Table 3), the statistical results of the average number of nouns per document indicate the prevalence of this feature in news (168.57), although this genre shares a very similar average with that of reviews (164.23), leaving behind children's tales with a relatively lower average (139.69). However, the different levels of prevalence between the genres become more obvious when comparing the values for proper nouns. Indeed, the average number of proper nouns per document shows the highest value in news (46.81), distanced to a great extent from the other two genres, with 33.68 for reviews and 22.84 for tales.

**Table 3.** Average POS and syntactic features per document.

| Features | News | Reviews | Tales |
|---|---|---|---|
| Nouns | 168.57 | 164.23 | 139.68 |
| Proper nouns | 46.81 | 33.68 | 22.84 |
| 1st-person pronoun | 3.63 | 17.02 | 10.73 |
| 2nd-person pronoun | 0.53 | 5.85 | 4.46 |
| 3rd-person pronoun | 11.98 | 9.94 | 34.08 |
| Adjectives | 32.22 | 46.36 | 38.35 |
| Adverbs | 23.84 | 48.81 | 52.24 |
| When-adverb | 0.99 | 1.85 | 3.43 |
| Wh-adverb | 1.90 | 3.52 | 6.07 |
| Present tense | 20.94 | 47.85 | 17.27 |
| Past tense | 29.05 | 18.26 | 61.42 |
| Future verb form | 1.84 | 1.86 | 1.99 |
| Infinitive verb form | 19.83 | 27.64 | 32.00 |
| Gerund verb form | 13.03 | 14.71 | 12.02 |
| Participle verb form | 18.73 | 14.64 | 17.91 |
| Predicative complement | 18.02 | 34.20 | 28.92 |
| Figures | 24.24 | 40.87 | 11.67 |

This widespread use of nouns and proper nouns by news was also identified by Dijk [53], who remarked on the high frequency of nouns repeatedly modified to make concretions about the elements being referred to.This author also stressed the importance of the phenomenon of nominalizations in this genre, which is due to its formal register, placing nouns ahead of the rest of the grammatical categories. Another approach is that of Biber and Conrad [26], who identified long sentences made up of one single verb and various nouns as one of the main features of news. Moreover, regarding proper nouns, their prevalence in news is also a consequence of their prototypical structure [53]. Here, the excerpts of information are presented from the most important to the most trivial facts, therefore repeating the same referents but always adding new details to the piece of news. In this way, those proper nouns involved in the events narrated are repeated several times so that readers know to whom or to what we are referring at any time.

Regarding reviews and children's tales, the lower prevalence of this syntactic feature is a reflection of the prototypical linguistic patterns of both genres. Reviews tend to include a mixture of grammatical classes in their texts due to their hybrid nature of narratives with descriptions and evaluations, whereas tales usually simplify most nouns and proper nouns to avoid making the story more complicated for children so that they can identify the main characters of the story effortlessly.

### 4.2.2. Personal Pronouns

Table 3 reflects how this syntactic feature helps to define children's tales to a larger extent, especially in the case of 3P pronouns, where tales have an average of 34.08 elements per document, in contrast with the 11.98 average of news and 9.94 of reviews. However, the latter genre shows, by far, the highest average of 1P pronouns per document, 17.02, while news remains the genre where personal pronouns have a more secondary role.

The use of personal pronouns, and concretely those belonging to 3P, is considered to be one of the most representative grammatical elements of those genres belonging to the narrative typology, and concretely to children's tales [25]. This prevalence is also confirmed by the previous results regarding *nouns and proper nouns*, as tales was the genre with the smallest presence of those elements, substituting specific nouns with 3P pronouns to not distract children from the narrative argument. Regarding news, the preference for 3P pronouns is mainly due to the impersonal style associated with news, as they are conceived as objective texts narrating in an impersonal way [52]. Regarding reviews, the higher presence of 1P and 2P pronouns comes from the involvement of users with their audience, as they often allude to the community directly to give them advice regarding the product or possible complications with the purchase [56]. These data help us to confirm the communicative purpose of reviews, as their main intention is to convey the personal opinion of the user regarding a product or service by showing a personal narrative focused on the writer's perspective [56], thus positively or negatively persuading the reader toward the reviewed product or service.

### 4.2.3. Adjectives and Adverbs/Wh-Adverbs

Firstly, regarding adjectives (Table 3), reviews stand out as the genre with the highest average of adjectives per document (46.36). Such relevance coincides with the communicative purpose of reviews, which is narrating users' personal experiences of a product or service as well as evaluating it [57]. Thus, adjectives become an essential feature to include relevant aspects related to the product to publish a detailed review. Secondly, children's tales show an average of 38.35 adjectives per document. Tales also use adjectives generally to include descriptions inside their narrative structure and to contextualize the situation in which the story takes place. However, the feature that arguably differentiates tales is the prevalence of adverbs over adjectives (Table 3). This genre has the highest average of adverbs per document (52.24), as well as when-adverb and wh-adverbs (3.43 and 6.07, respectively). Consequently, adverbs—all their types—are quite a distinctive feature of the textual genre of children's tales. Indeed, Smith [3] remarked on the role of adverbs, and temporal ones within them, when providing dynamism to any narrative to make the events progress sequentially. Therefore, not only temporal adverbs, but also wh-adverbs, help to locate the main events of the story in order to identify the different elements that it comprises. Adverbs such as *when*, *why*, *how*, and *where* serve to introduce the main aspects regarding the development of the story, being even more relevant in tales directed to children to guide them in their understanding process. Regarding news, both features have a lower presence throughout its corpus (with an average of 32.22 adjectives and 23.84 adverbs per document). This is mainly due to the formal and objective style of this genre, where texts are conceived as external events that have to be told without including messages that may lead the reader to believe the events in a biased manner. Certainly, this preference for an objective style matches the communicative purpose of news, which is informing of current events without delving into personal opinions or insights.

### 4.2.4. Verbal Tenses

Verbs are traditionally considered to be one of the fundamental linguistic elements linked to narrative typology [4], as they represent the actions that advance the story to fulfill its expected dynamism. Table 3 shows how children's tales have a clear preference for past verbal forms, with an average of 61.42 past tenses per document. Meanwhile, news tends to include a mix of present and past forms, according to their average presence per document (20.94 and 29.05, respectively), while reviews prefer a greater use of present verbal forms, with 47.85 average elements per document. On the one hand, the preference of children's tales for past tenses agrees with the opinions of many authors that highlighted the use of this verbal tense to chronologically narrate the events occurring in the story [52,58]. On the other hand, the mix of present and past tenses in news is due to the fact that this genre generally uses past tenses to describe the occurring events and, according to Zheng [59], it also makes use of the present tense, especially for the headlines and subtitles which lead to the development of the story. Finally, the reverse trend of reviews arises from the general use made by this genre of the present tense as a mechanism to generalize users' statements and therefore declare "universal truths" [35]. As a consequence, past tenses are only used for narrating past actions related to the purchase of the product or to justify why the user needed or wanted such a product. Given these results, the almost absent verbal form is the future, reasonably because narratives are mainly used to describe a sequence of actions that progressively developed, without advancing what may happen in the future.

### 4.2.5. Nonfinite Verb Forms

Nonfinite forms can be equally observed in each of the three genres studied, depending on the particular verb form we focus on. Regarding children's tales, this genre stands out for a widespread use of the infinitive form, whereas reviews have the highest average of gerund elements. Meanwhile, news has the highest presence of participle forms of the present analysis.Table 3 shows that the three genres coincide in their highest prevalence of infinitive aspects over the other two types, with an average of 32.00 elements per document in tales, 27.64 in reviews, and 19.83 in news. Indeed, nonfinite verbs differ from the rest of the verb forms because they cannot be the verb head of a sentence and, therefore, they usually depend on another verb [60]. As a consequence, the highest prevalence given to the infinitive form in the statistical values is mainly due to the wide range of functions that this verbal form has, as it can work as a noun, a subject, a complement, an object, or a modifier, among others [61]. Therefore, it can be easily used to fulfill any of the aforementioned functions, and it is a simple form that can be easily recognized by any reader, regardless of their age. Regarding the participle form, this aspect shows the second-highest prevalence in the genres. Participles, according to Herrero Salas [61], mainly work as either adjectives or adverbs in the sentence, consequently having the function of modifiers. In this way, the participle verb form helps the construction of the story by providing further details about the descriptions and actions narrated in the events. Finally, regarding the gerund aspect, it shares some of its functions with the infinitive aspect, as it can also work as the subject, object, or complement of the sentence, due to its noun condition. However, it is not as mentioned in the different genres as the other types of verb aspects except for reviews, which opts for a balance between this aspect and the participle, arguably due to the variety of writing styles that reviewers can use when narrating their personal experiences.

### 4.2.6. Predicative Complement

The predicative complement is mainly used to include further information about a noun, generally through predicate adjectives or nouns following a copulative verb (such as *to be*, *to seem*, *to appear*, etc.). Therefore, analyzing the prevalence of this complement seems interesting to infer to what extent narratives include more descriptive and contextual passages in their stories. In line with this, Table 3 reflects how the predicative complement has a clear trend in reviews given its linguistic function (with 34.20 average elements per

document), closely followed by tales, showing 28.92 average elements. Finally, news falls behind with 18.02 predicative complements per document.

This preference of reviews may be due to the motivation of the reviewer, who voluntarily writes these comments to share experiences and help others, either by criticizing or praising the product. In any case, the user normally tries to include as much information as possible, both in the evaluation part (e.g., "*the laptop is incredibly light*"),and in the product description (e.g., "*the room was not insulated*") [62]. Therefore, users of online reviews will generally make use of this feature, depending on the amount of information they want to include in the narration of their experience and the evaluation of the product. In the case of children's tales, most of them are characterized by the use of evaluative and attitudinal lexis to instill moral or social values in the young child [52], consequently making use of the predicative complement to include that evaluative aspect to the narration of the story. Regarding news, the communicative purposes are quite the opposite in this genre, as the information needs to be told in a formal and objective way, without letting a particular attitude arise towards any topic, consequently showing a lower prevalence of this feature.

### 4.2.7. Figures

Even though this feature could be considered as one of the "shallow" elements included in this research, we added figures to the syntactic level of analysis as we took into account numerical values and words expressing figures, therefore needing a preprocessing to be identified as such.

Table 3 shows the statistical compilation of the prevalence of figures in the three corpora. We observe that reviews have the highest average of figures per document: 40.87. This widespread use of figures by reviews may be due to the varied linguistic strategies employed by users to narrate their experiences, as well as the multiple types of reviews that this study includes. Indeed, the reviews gathered for the present corpora range from products and films to hotel reviews, among others, and in those examples, figures take a leading role in order to rate the product (e.g., "*VERY Stupid Question . . . 5 Stars*"), or to give details about the products (e.g., "*That's $20.40 for a breakfast that has an intrinsic value of about five bucks*"). Regarding news, this genre has 24.24 average figures per document, in line with the general use of this feature, depending on the topic of each piece of news. Numbers can be present thanks to their informative value, although becoming more relevant in specific news regarding topics such as economics or international trade. Finally, tales have an average of 11.67 figures per document, consequently not becoming a fundamental feature of the genre in comparison to the others.

### 4.3. Semantic Features

The third level of linguistic analysis comprises the different components that semantically affect the overall meaning of the sentences, delimiting the interpretation of the elements mentioned in each narrative.

### 4.3.1. Events

A first look at Table 4 already reflects the importance that events have for the course of the narrative, being one of the cornerstones of the story used to advance from the introduction to the final resolution [3]. This general pattern is reflected in the average number of events per document, with an average of 66.62 in reviews, 68.58 in news, and 97.62 in tales. These results reflect the main role of events to fulfill the communicative purpose of any narrative, which is to develop a series of events and states in a dynamic and straightforward way.

**Table 4.** Average semantic features per document.

| Features | News | Reviews | Tales |
|---|---|---|---|
| Events (general) | 68.58 | 66.62 | 97.62 |
| Aspectual event | 0.69 | 0.46 | 0.82 |
| Intensional action event | 4.18 | 2.85 | 6.41 |
| Intensional state event | 3.35 | 6.49 | 3.89 |
| Occurrence event | 49.77 | 52.29 | 77.51 |
| Perception event | 0.22 | 0.64 | 0.80 |
| Reporting event | 8.69 | 1.66 | 5.37 |
| State event | 1.69 | 2.33 | 2.81 |
| Time links general (Tlinks) | 381.44 | 240.80 | 592.92 |
| Before Tlink | 46.55 | 29.82 | 104.08 |
| After Tlink | 36.90 | 17.12 | 67.02 |
| Includes Tlink | 1.79 | 0.99 | 1.74 |
| Is included Tlink | 15.58 | 5.79 | 9.94 |
| Simultaneous Tlink | 1.93 | 1.66 | 2.77 |
| Vague Tlink | 278.68 | 185.43 | 407.38 |
| Parenthesis sentences | 0.00 | 3.91 | 0.00 |
| Interrogative sentences | 0.21 | 1.75 | 2.42 |
| Exclamatory sentences | 0.06 | 2.21 | 2.85 |
| Nominal subject | 54.09 | 76.02 | 91.69 |
| Direct object | 31.37 | 38.44 | 42.86 |
| Named entity length 1 (NE 1) | 31.12 | 22.21 | 19.79 |
| NE length 2 | 9.97 | 8.46 | 2.34 |
| NE length 3 | 3.46 | 2.00 | 0.43 |
| NE length 4 | 1.30 | 0.66 | 0.12 |
| NE length 5 or more | 0.96 | 0.42 | 0.16 |

Focusing now on the types of events identified in the present analysis, which are extracted from the TimeML guidelines in Saurí et al. [63], Table 4 illustrates a general preference for events of the category *ocurrence*, with an average number of elements per document of 49.77 for news, 52.29 for reviews, and 77.51 for tales. This is due to the varied amount of events that this typology comprises, as *occurrence* events include all the many other kinds of events describing something that happens or occurs in the world, setting aside the more specific types of events comprising *perception* or *state* events, among others. Therefore, when omitting the *occurrence* type to more concretely measure the prevalence of the rest of the events, we observe that the prevalence in each corpus is way more varied.

Regarding news, Table 4 indicates that this genre is defined by *reporting events*, with an average of 8.69 elements per document. *Reporting events* describe the action of a person or an organization declaring something, narrating, or informing about an event, etc. [63], which coincides with the main communicative purpose of news [26]. Regarding reviews, their communicative purpose is also responsible for the general use of events of the *intensional state* type, as reviews are intended to give the user's opinions regarding a product or service [64]. Indeed, this genre has an average of 6.49 *intensional state* events per document. Finally, children's tales show a more balanced frequency of use between *reporting* events, with an average of 5.37 per document, and *intensional action* events, with an average of 6.41. This combination derives also from the communicative purpose of the genre, which is to narrate a sequential series of actions, introducing event arguments describing situations from which we can infer further information related to that event [63].

### 4.3.2. Time Links

According to Saurí et al. [63], a temporal link represents the temporal relation holding between events, times, or between an event and a time (e.g., *before, after, simultaneous*). Therefore, this feature can reflect a lot of semantic information in the document, indicating the relation between the different events or times that appear in the course of the story. Indeed, Table 4 already indicates the high values that the average number of TLinks have

in general, with an average of 240.80 in reviews documents, 381.44 in news, and 592.92 in tales, being the highest average in the entire study.

As for the particular types of TLinks, Table 4 shows that the TLink with the highest prevalence in each of the three genres is *vague*. This feature, which indicates that the relation between two possible events or two times or a mix of both is not clear, or it does not exist, has an average of 407.38 per document in tales, 278.68 in news, and 185.43 in reviews. Apart from *vague*, we observe that the following TLinks with the highest representation in each genre are *before* and *after*, given that both TLinks refer to the same temporal relation, as one event or time can occur before or after another, only changing their direction. This coincides with the prototypical progress of a narrative, as it generally develops the events chronologically to narrate actions in an orderly manner [65]. Indeed, TLinks are especially relevant in children's tales, where the average number of *before* and *after* time relations per document is 104.08 and 67.02, respectively. Such results agree with Labov and Waletzky's [4] classification of the basic units that any narrative should have, which includes temporal expressions. In contrast, news shows an average of 46.55 *before* TLinks per document and 36.90 *after* TLinks. These different results come from the prototypical structure of news, where the most important information is narrated first, instead of following the chronological order. Therefore, and according to Dijk [53], the associated temporal relation of *before/after*, which is essential in storytelling, may be replaced by functional relations of specification. Regarding reviews, the values indicate even lower averages, with 29.82 for *before* Tlinks and 17.12 for *after* TLinks, as reviews include a wider variety of textual typologies (mixing narratives with descriptions or evaluations). Nevertheless, they still have a remarkable role in the genre, as the most highly narrative accounts tend to include forms of temporal deictic anchoring [56] by using such TLinks.

### 4.3.3. Named Entity Length

In order to analyze to what extent this feature influences each genre, we have to consider the existing types of named entities (NEs), these being expressions referring to locations, organizations, or people, among others. For the present research, NEs of different lengths were considered, calculating their presence depending on the number of words each NE comprises, with NEs that range from one up to five words, as shown in Table 4. The results show that NEs made up of one word have the greatest average of 31.12 for news, whereas reviews have an average of 22.21 and tales of 19.79, being the type of NE with the highest representation in the three corpora. Indeed, this feature matches the "top-to-bottom" structure of news, by which the same elements are largely repeated throughout the story, but each time focusing on different details depending on their relevance, therefore showing the same NEs in different parts of the document. Regarding reviews, users have as their main purpose to narrate their personal experience with a product or service, therefore including NEs referring to the different brands that the products belong to, or to countries or cities where their experience took place. Lastly, the target audience linked to the genre of children's tales are generally children, which limits the variety of NEs in the narrative to not hamper children's understanding of the story, preferring NEs of one element to help them to memorize the different names [66]. Consequently, Table 4 indicates how the number of NEs made up of two elements plummets in tales in comparison to the other two genres.

### 4.3.4. Parenthesis, Interrogative, and Exclamatory Sentences

One of the reasons parentheses are commonly used in most types of narratives is their linguistic function, as they help to clarify and extend the information when needed. At the same time, the main function of certain special marks such as question and exclamation marks is to enhance the emotional tone of a sentence [67]. These facts motivated our interest in analyzing the presence of these orthographic elements as exponents of semantic meaning within the discourse.

According to Table 4, we observe that the three types of sentences do not seem to be a basic feature for news, as each of them has some of the lowest average number of elements

per document, being 0.21 for interrogative sentences, 0.06 for exclamatory, and no results in the case of parentheses. This is a reflection of one of the main communicative intentions of the genre, as news normally narrates daily events in an objective way [10], developing the facts without providing any personal opinion or emotion. Moreover, parentheses can be used in narrative to add secondary details to the story, which highly contrasts with the conventions of news, where it is presumed that all the information included is equally important, without having to add or to clarify previous data.

Regarding the other two genres, children's tales show a higher average of both interrogative and exclamatory sentences per document (2.42 and 2.85, respectively), setting aside parentheses with no examples in the genre. Consequently, reviews stay in second place for the average number of interrogative and exclamatory sentences (1.75 and 2.21, respectively), but surprisingly exceeds the other genres in the presence of parentheses, with an average of 3.91 per document. These different values in tales and reviews are also due to the communicative purposes of the two genres. On the one hand, it is of great importance that the actions told in tales are explained dynamically [68] to maintain children's attention so that they can follow the course of the story. Such conditions are successfully accomplished by using both exclamatory and interrogative sentences, as the former help to draw readers attention [69] as well as to announce coming events [70], and the latter momentarily interrupt the thread of the story [52] so that children can become involved in the story by asking themselves the same questions made by the characters. On the other hand, the use of parentheses can interfere with the dynamism and flow of the narrative, perhaps adding exceeding details, given that children's ability to internalize the information of a sentence is much more limited. On the contrary we have reviews, which mainly stand out for their use of parentheses, as they help users to add specifications about the product, directly address the community of users with an informal register, or express their feelings—by also including exclamatory and interrogative sentences—which coincides with Vásquez's [56] analysis on the linguistic mechanisms that are generally used in reviews.

### 4.3.5. Subject and Object Dependency Relations

It can be observed that although this linguistic feature is proportionally present in the three genres, it could be a more representative characteristic of children's tales, while news and reviews lag somewhat behind, but with very similar results. This is given by the results indicated in Table 4, where the three genres have quite high averages of elements per document, with an average number of nominal subjects of 91.69 in tales, 76.02 in reviews, and 54.09 in news. Arguably, these values indicate that narratives generally prefer the use of nominal subjects to mention the different entities, characters, or people involved in their respective stories clearly. Such preference coincides with one of the fundamental linguistic features that any narrative must include, which is simple subjects [4]. However, tales show, by far, the highest average, which may be mainly due to the particularities of the genre given its target audience, as children need to have precise referents reflected by those nominal subjects. Regarding the presence of direct objects in the three genres, the statistical values of news and reviews come close to each other, with an average number of 31.37 direct objects per document in news, and 38.44 in reviews. Just as before, children's tales also show the highest value for this feature, with an average of 42.86, which also coincides with another of the fundamental linguistic features of narratives according, again, to Labov and Waletzky [4], which are various types of complement structures (including direct objects).

### 4.4. Discourse-Related Features

Discourse features can give many interesting insights into the patterns that texts may follow for their structural organization or topic development, which generally determine the readability level that each text shows [71].

### 4.4.1. Coreference

Coreference can represent the implicit relations between sentences [72], therefore becoming of great importance for defining the overall structure of the discourse belonging to any genre. According to Table 5, children's tales is the genre with the highest average of coreference chains per document, 18.33. This clear preference derives from one of the fundamental linguistic premises of this genre, which is narrating stories with concise and simplified language for the total understanding of the child. Consequently, coreference helps the target audience to follow the development of the story, as well as to recognize which character performs each action, referring back to them each time that an event is described [73]. These concrete characteristics of narrative also coincide with the highest average of *maximal chains* per document that tales show, 10.36—which in our present corpora means a total of three referents for the same element—as they help to guide the child throughout the story, identifying the different ways of referring to the same character. Then, news has the second-highest average number of coreference chains per document, 16.47, as well as the second-highest average number of *maximal chains* per document, 7.94. The use of coreferences in news can be due to the "top-to-bottom" structure of the genre, where the distribution of the actions repeats the different excerpts of information from the most important to the most trivial ones, although always adding further information [73]. By repeating such information, the references to those entities also increase, in order to follow the events and the people involved in them. Finally, reviews lag behind with an average of 15.25 coreference chains per document and 7.60 *maximal chains*, which may be a consequence of the different communicative purposes they can fulfill (e.g., evaluating the product, giving advice to possible customers, narrating the purchasing experience, etc.). Thus, reviews can change the topic—and, therefore, the referents—more often than in the other genres, consequently showing fewer coreference chains in its corpus.

**Table 5.** Average discourse features per document.

| Features | News | Reviews | Tales |
|---|---|---|---|
| Coreference chains | 16.47 | 15.25 | 18.33 |
| Maximal coreference chains | 7.94 | 7.60 | 10.36 |
| Discourse markers (DMs) | 13.05 | 20.74 | 22.61 |
| Time expressions general (timex) | 12.06 | 6.23 | 7.39 |
| Date timex | 6.83 | 2.66 | 3.07 |
| Time timex | 0.74 | 0.66 | 1.29 |
| Duration timex | 4.21 | 2.65 | 2.71 |
| Set timex | 0.28 | 0.26 | 0.33 |
| Quotation marks | 6.23 | 0.10 | 12.32 |

### 4.4.2. Discourse Markers

In order to identify and analyze the different types of discourse markers (DMs) found in the three genres, the *lexicon of prototypical discourse markers* in Alemany [50] was used as the classification guideline. According to such study, discourse markers are characterized by their structural (i.e., *continuation* or *elaboration*) and semantic (i.e., *revision*, *cause*, *equality*, or *context*) meanings, which helps to develop a deeper analysis on the different particularities of each DM type. In the present study, we analyzed the distribution of each DM type, showing that the three genres have a clear preference for *elaboration* DMs regarding the structural meaning, as well as for *context* DMs when focusing on the semantic meaning.

Furthermore, Table 5 indicates that tales have the highest average of discourse markers per document, 22.61. Meanwhile, reviews stay in the second position with an average of 20.74, and, finally, news shows an average of 13.05 DMs per document. The statistical values reflected in tales are due to the limiting nature of discourse markers, which help to focus the interpretation of the sentence that follows such a DM by restricting its meaning [74] depending on the purpose of the message. Therefore, discourse markers help children understand the meaning of the utterances included after them, guiding them through

the resolution of the story. Regarding reviews, the study of Crotts et al. [75] already demonstrated how discourse markers were generally used in TripAdvisor comments to express users' opinions on the experiences purchased through this website, as well as to advise other users of the same community. Indeed, DMs can also serve as discourse units that structure social interaction among the different participants at different levels [76]. As a consequence, given the "hybrid" nature of reviews, which are typically associated with both oral and written linguistic genres, many users often make use of them to organize their discourse. The opposite conventions are linked to news, consequently due to the written nature of this genre, which tends to avoid any example of oral language. Derived from this, although this feature is generally used in news to connect the different actions that occur and restrict the meaning of the sentences, they are not the most prevalent feature in this genre.

### 4.4.3. Time Expressions

Following the annotation guidelines included in Saurí et al. [63], four different types of time expressions (timex) were considered in this paper: *date* (e.g., "*on Wednesday*"), *duration* (e.g., "*for months*"), *time* (e.g., "*at night*"), and *set* (e.g., "*weekdays*"). Time expressions in Table 5 present some of the lowest statistical values of this research, but still have some presence in news, with 12.06 average elements per document. Regarding children's tales and reviews, these genres reflect similar results, with 7.39 and 6.23 average timex per document in each corpus. The higher trend of timex in news comes as a consequence of the particular structure by which events are told in this genre, as news generally move through the different events back and forward, depending on the importance that lies in each information excerpt. News also stands out for the widespread use of the particular type of timex *date*, with an average of 6.83 elements per document. Such prevalence is also due to the prototypical structure of the genre, by which the journalist specifies the concrete dates in which the information occurred to set the timeline of the events. Regarding reviews, Table 5 shows how this genre has very similar values for the average number of *date* and *duration* timex per document, 2.66 and 2.65, respectively. These similar results for such timex may be due to the unpredictable nature of reviews, as the text can change easily depending on the aspects that each online user wants to remark on regarding the product (e.g., to express their gratitude or disappointment on the estimated shipping time or to narrate their experience, specifying how much time they spent in a hotel or on concrete events that took place during their stay). Regarding tales, this genre has an average number of *date* and *duration* timex of 3.07 and 2.71, respectively, as well as the highest average of *time* elements per document, 1.29. This slight preference of tales for *time* timex can be due to the prototypical structure of the genre, where the events are narrated chronologically within short periods, especially for tales aimed at small children [77]. In this way, the time interval in which the actions are developed is more concrete, therefore becoming easier for children to follow the timeline of the events.

### 4.4.4. Quotation Marks

Although this feature could be considered part of the "shallow" linguistic level at first glance, we decided to analyze it as part of the discourse level, given its influence on the discursive interpretation of the texts. This is due to its fundamental textual function, as it differentiates those textual passages written in direct speech from the external narrator's perspective, as well as quoting messages uttered by any of the characters of the story.

The two genres with remarkable averages of the number of quotation marks per document in Table 5 are both news and tales, with an average of 6.23 and 12.32, respectively, which highly contrasts with the 0.10 average of reviews. Regarding tales, this linguistic element is a basic resource for any narrative that introduces passages in direct style showing dialogues between the characters of the story. Regarding news, the use of this feature helps the genre differentiate between the distinct levels of discourse, therefore separating the narrator's perspective and that of the people involved in the events. Moreover, this feature

is used in news as one of the fundamental linguistic mechanisms of the journalist in order to write an unbiased and impartial piece of news [53]. Finally, reviews stand out as the genre with one of the lowest averages per document of the entire study, 0.10. This value is in line with the prototypical characteristics generally linked to this genre, as reviews are usually written from the first-person perspective [56,78], so that online users can narrate their experience with products by including all the relevant information they consider. Consequently, quotation marks expressing the messages of people other than those users are hardly ever found in these texts.

## 5. Overall and Final Remarks

Our aim with this research was to find out to what extent specific linguistic features are determined by the communicative purpose generally linked to each genre, and to extract linguistic patterns that could prove the inclusion of well-distinguished genres such as news, tales, or reviews inside the textual typology of narrative. After the multilevel linguistic discourse analysis, the heatmap shown in Figure 1 serves as a guide to easily find those defining features in each linguistic genre, and to show how the features with the highest prevalence in each genre are actually linked to their respective communicative purposes. To clearly distinguish the linguistic patterns extracted in news, reviews, and tales, those features with the highest prevalence are colored in red, whereas the second-highest predominant features are indicated in blue. Meanwhile, those features with the smallest presence in each genre correspond to the light blue fragments of the heatmap.

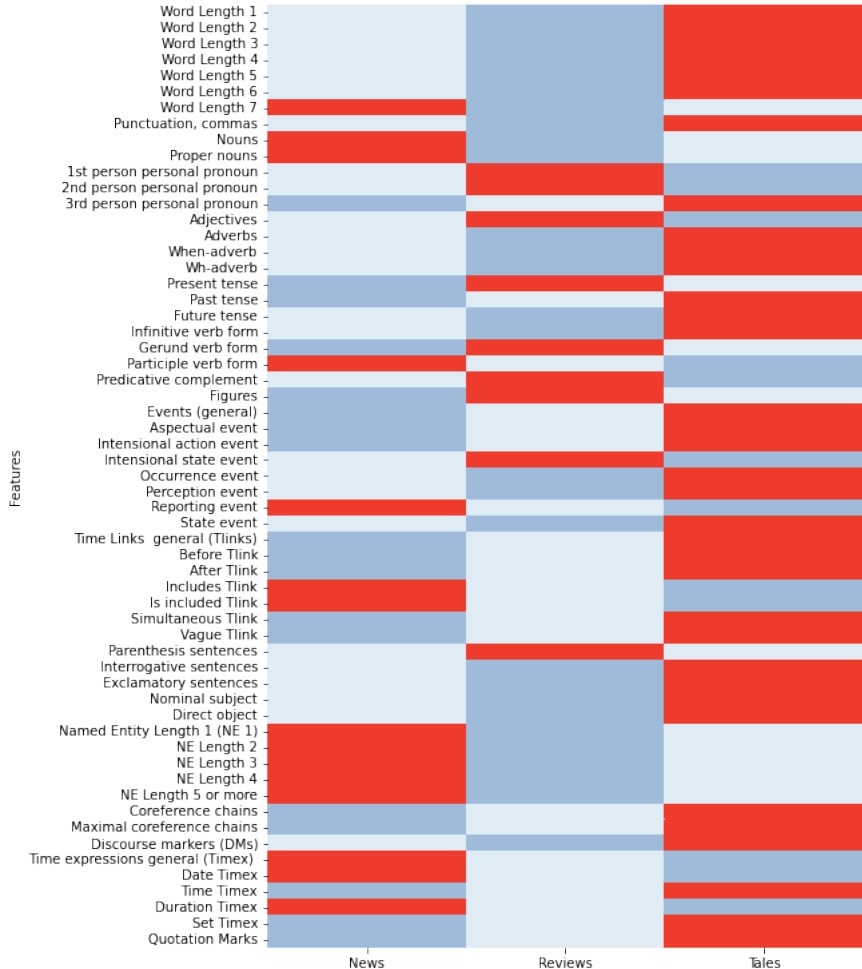

**Figure 1.** Heatmap showing the prevalence of features per genre.

Regarding news, we observed that some of the features with the highest prevalence are those referring to the identification of people or places (*nouns and proper nouns*, *proper noun phrases*, and *named entity length*), as well as the widespread use of words with a minimum of seven letters. Recalling the main communicative purpose of the genre, which is to inform readers about daily events occurring in the world in the most accurate way, this helps us to confirm the abundance of *reporting event* verbs in the genre. In addition, given the statistic results, it seems clear that news is arguably focused on the distinction of the people involved in those events and choosing the most appropriate words for the narration. Moreover, news tends to focus on the disposition of the events according to its "top-to-bottom" structure; therefore, a general use of *time expressions* prevails to arrange the information excerpts appropriately.Consequently, we could argue that this genre actually makes use of such features to accomplish its predominant communicative purpose, as it ensures that the events are accurately told by identifying the actions and their main participants inside the timeline of the piece of news.

Regarding reviews, the most prevalent features in the genre are also arguably linked to their principal communicative purpose. It is true that this genre, as a type of computer-mediated communication, is more prone to have a mix of communicative purposes depending on the own intentions of each online user. Indeed, in the same review, users focus on the product description and on adding a narrative component to the discourse by telling their personal stories regarding such products. Consequently, reviews generally show features such as *parenthesis sentences*, *present-tense* verbs, and *figures*, in order to add as many relevant details on the reviewed product as possible, as well as *adjectives* and *first- and second-person pronouns* in order to persuade the rest of the online community. Arguably, most of the online reviews analyzed in the present study include such narrative component by using the linguistic features that shape those reviews in order to also accomplish their fundamental communicative purpose of evaluating a product. Hence, online users tend to emphasize their opinion of the product by creating personal narratives on their experience with such products.

Finally, tales generally have the communicative purpose of entertaining the readers by making them feel involved throughout the story. However, in the particular case of children's tales, such communicative purpose is arguably more restrictive given the average age of the target audience. With this in mind, we observe that this genre could also fulfill its main purpose with specific linguistic features aimed at the entertainment of children. For example, the general use of *short words* for children's easy understanding as well as *commas* or *coreference chains* helps them to follow the narrative rhythm and the actions performed by each character. Other features that stand out in the genre are also *events* and *time links*, therefore focusing on the development of the actions and their chronological progress. Additionally, features such as *third-person pronouns* and *past verbal tenses* show some of the fundamental elements that every narrative discourse should include according to traditional literary theory. By means of all these features, tales tend to help children to feel more involved in the story and therefore entertain them throughout the narrative, consequently justifying the use of such linguistic features to complete their communicative purpose.

## 6. Conclusions and Future Work

In this paper we presented a data-based approach to analyze and detect genre-specific and prevalent patterns that can be found in the structure and discourse of narrative texts belonging to three different genres. We were interested in extracting and discovering several linguistic phenomena of the narrative discourse at several levels of analysis, and in which elements—among those that can be accounted through automatic tools—may relate to the structure of stories. More concretely, a set of 59 linguistic features belonging to four different linguistic levels (shallow, syntactic, semantic, and discourse-related features) and containing information at the document level were the most appropriate for this

research. The corpora gathered for the multilevel analysis were heterogeneous and came from different sources, covering the genres of news, reviews, and children's tales in English.

The multilevel discourse analysis performed suggests that a group of heterogeneous linguistic features provides a more accurate representation of the peculiarities that differentiate each genre, presumably because the particular nature of a genre is hardly determined by a unique linguistic level, as it can be appreciated in the different linguistic levels of features that shape each narrative genre according to the heatmap. Additionally, the findings also indicate that many of such linguistic patterns may be defined to some extent by the communicative purpose that each genre reflects, given the direct connection between their appearance in the genre and their respective communicative purposes. This relation could also be justified by means of the heatmap illustrating the prevalence of the features in each genre, as those linguistic patterns with higher prevalence in one particular genre do not prevail as much in the rest of the textual genres, therefore confirming the validity of the two research questions that were proposed for this study.

As future work, we plan to extend the analysis to other genres and languages, since similar linguistic tools can be found for this purpose. With the parallel data coming from this stage, a deeper multilevel study of the dynamics established between the elements of narrative discourse and the actions that occur to the characters at certain moments will be possible.

For future publications, it would be interesting to consider other aspects regarding the linguistic patterns of the genres included in the study to provide further conclusions. Given that texts can show a variety of communicative purposes to fulfill, we could also extract additional pragmatic patterns of the discourse by analyzing how each communicative purpose influences the information included in a text of a particular genre. For instance, a children's tale can either entertain or instill moral values in the child, and a review can be used to persuade an online community or to describe a product, which perhaps could have an influence on which linguistic features may attain a higher prevalence in the same genre depending on the communicative purpose to achieve.

**Author Contributions:** Conceptualization, M.M.M., M.V., E.L. and A.S.C.; methodology, M.M.M., M.V., E.L. and A.S.C.; software, M.V.; formal analysis, M.M.M. and M.V.; investigation, M.M.M., M.V., E.L. and A.S.C.; resources, M.M.M. and M.V.; data curation, M.V.; writing—original draft preparation, M.M.M. and M.V.; writing—review and editing, E.L. and A.S.C.; supervision, E.L. and A.S.C.; project administration, A.S.C. and E.L.; funding acquisition, E.L. and A.S.C. All authors have read and agreed to the published version of the manuscript.

**Funding:** This research work is part of the R&D project "PID2021-123956OB-I00", funded by MCIN/AEI/10.13039/501100011033/ and by "ERDF A way of making Europe". Moreover, it was also partially funded by the project "CLEAR.TEXT: Enhancing the modernization public sector organizations by deploying natural language processing to make their digital content CLEARER to those with cognitive disabilities" (TED2021-130707B-I00), by the Generalitat Valenciana through the project "NL4DISMIS: Natural Language Technologies for dealing with dis- and misinformation" with grant reference CIPROM/2021/21, and finally by the European Commission ICT COST Action "Multi-task, Multilingual, Multi-modal Language Generation" (CA18231).

**Institutional Review Board Statement:** Not applicable.

**Informed Consent Statement:** Not applicable.

**Data Availability Statement:** The data presented in this study regarding the SFU reviews corpus [42] and the Lobo and Matos corpus of fairy tales [40] are openly available at https://www.sfu.ca/~mtaboada/SFU_Review_Corpus.html (accessed on 16 September 2021) and https://www.hlt.inesc-id.pt/w/Fairy_tale_corpus (accessed on 10 July 2021). However, the *Bedtime Stories* corpus of tales [39] and the DUC news corpus [41] are only available upon request to the corresponding authors through their respective websites: https://freestoriesforkids.com/ (accessed on 24 March 2021) and https://duc.nist.gov/data.html (accessed on 6 December 2020). The code used to conduct the processing of the aforementioned corpora is openly available here: https://github.com/mvmUa/narrative_discourse_patterns, accessed on 15 November 2022.

**Conflicts of Interest:** The authors declare no conflict of interest.

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
