# Peer review of "Extracting Narrative Patterns in Different Textual Genres: A Multilevel Feature Discourse Analysis"

_information, doi:10.3390/info14010028_

Round 1

Reviewer 1 Report

l.179 While it is clear for this reviewer that the data is in English, it may be not so clear for the readers. Please specify the language for them.

l.201 Why use a multilingual tool if the data is in English? Is it better than language-specific tools?

Table 2: Is there a reason for not addressing sentence length? Please explain if possible.

Author Response

Point 1. 

l.179 While it is clear for this reviewer that the data is in English, it may be not so clear for the readers. Please specify the language for them.

Response for Point 1.

In order to clarify that our data is in English, we added "In line with this, several corpora were collected from various sources in English to compile [...]" in line 178.

Point 2.

l.201 Why use a multilingual tool if the data is in English? Is it better than language-specific tools?

Response for Point 2.

Freeling was the tool we were most used to work with in order to extract the features that we were interested in for our research. Moreover, as we plan to extend our research on this topic to other languages, we thought it would be easier to work with the same tool throughout all our research on Information Extraction. 

Point 3.

Table 2: Is there a reason for not addressing sentence length? Please explain if possible.

Response for Point 3.

We believed that the "sentence length" feature was not going to provide with many remarkable insights because in Table 1 we can already observe that the average number of words per sentence is quite similar in each genre. Therefore, we could not infer any preferences for a certain sentence length in a particular genre. That is the main reason why we preferred to focus on other features that showed more differentiated values to provide with clearer conclusions. 

Reviewer 2 Report

This study should not be limited to english language as it doesn't required large language resources. It could be easily extended to all latin languages in Freeling. In the specific case of french language it could help IR engines for children and schools like Qwant Junior. Theses engines can only use terms in queries and document content (no user history) and heavily rely on linguistic features and lexicons. 

Author Response

Point 1.

This study should not be limited to english language as it doesn't required large language resources. It could be easily extended to all latin languages in Freeling. In the specific case of french language it could help IR engines for children and schools like Qwant Junior. Theses engines can only use terms in queries and document content (no user history) and heavily rely on linguistic features and lexicons. 

Response for Point 1. 

We appreciate this comment because we plan to extend our research further in other languages so that we can explore the potential of this type of discourse analysis in the scope of information extraction on digital tools. We were unaware of Qwant Junior search engine, but it would be really interesting to check how the types of features we extracted may help this search engine to find the best results depending on their social context, as the results are directed to children straightaway.

Reviewer 3 Report

Comments and suggestions for authors

This is a well-written, logically organized paper addressing a niche within computational narratology. The authors pinpointed research questions that help develop the research field. Their research design appears sound. The results are presented clearly in both textual and visual forms, e.g. using heatmaps and tabulating values.  The paper is well referenced. I have only one minor but important issue to raise.

Minor issues

1. Unacknowledged overlap with previous paper. I understand that the research questions differ and that the datasets are similar although not identical.

Language issues

1. There are a few non-intrusive language errors, which should be addressed before submitting camera-ready copy, e.g.

Line 6    To accomplish so - - > To accomplish this

Author Response

Point 1. Minor issue.

  1. Unacknowledged overlap with previous paper. I understand that the research questions differ and that the datasets are similar although not identical.

Response for Point 1.

On the one hand, we believe that our previous paper was arguably more focused on the different Machine Learning techniques used to compare the performance of the algorithms chosen to classify those linguistic features, which are distributed in different groups to check which blend showed better results when testing their relevance in each genre. On the other hand, this second paper is more linked to the information extraction task by analysing the linguistic factors that make each feature be linked to a particular genre within the narrative typology to find out to which extent those features vary depending on the communicative purpose of each discourse.

Point 2. Language issues.

  1. There are a few non-intrusive language errors, which should be addressed before submitting camera-ready copy, e.g.

Line 6    To accomplish so - - > To accomplish this

Response for Point 2.

Thanks for pinpointing such language errors. We have revised the whole manuscript and have changed “To accomplish so” for “To accomplish this” in line 6, as well as other minor errors we did not noticed before.